# Formulated Curcumin Prevents Paclitaxel-Induced Peripheral Neuropathy through Reduction in Neuroinflammation by Modulation of α7 Nicotinic Acetylcholine Receptors

**DOI:** 10.3390/pharmaceutics14061296

**Published:** 2022-06-17

**Authors:** Martial Caillaud, Danielle Thompson, Wisam Toma, Alyssa White, Jared Mann, Jane L. Roberts, John W. Bigbee, David A. Gewirtz, M. Imad Damaj

**Affiliations:** 1Department of Pharmacology and Toxicology, Translational Research Initiative for Pain and Neuropathy, Medical College of Virginia Campus, Virginia Commonwealth University, Richmond, VA 23284, USA; thompsondc3@vcu.edu (D.T.); wisam.toma@vcuhealth.org (W.T.); awhite1357@gmail.com (A.W.); jared.mann@vcuhealth.org (J.M.); jane.roberts@vcuhealth.org (J.L.R.); m.damaj@vcuhealth.org (M.I.D.); 2Nantes Université, INSERM, The Enteric Nervous System in Gut and Brain Disorders, IMAD, F-44000 Nantes, France; 3Department of Anatomy and Neurobiology, School of Medicine, Virginia Commonwealth University, Richmond, VA 23284, USA; john.bigbee@vcuhealth.org; 4Departments of Pharmacology, Toxicology and Medicine and Massey Cancer Center, Virginia Common-Wealth University, Richmond, VA 23284, USA; david.gewirtz@vcuhealth.org

**Keywords:** curcumin, chemotherapy, peripheral neuropathy, neuroinflammation, α7 nACh receptors

## Abstract

Paclitaxel is widely used in the treatment of various types of solid malignancies. Paclitaxel-induced peripheral neuropathy (PIPN) is often characterized by burning pain, cold, and mechanical allodynia in patients. Currently, specific pharmacological treatments against PIPN are lacking. Curcumin, a polyphenol of Curcuma longa, shows antioxidant, anti-inflammatory, and neuroprotective effects and has recently shown efficacy in the mitigation of various peripheral neuropathies. Here, we tested, for the first time, the therapeutic effect of 1.5% dietary curcumin and Meriva (a lecithin formulation of curcumin) in preventing the development of PIPN in C57BL/6J mice. Curcumin or Meriva treatment was initiated one week before injection of paclitaxel and continued throughout the study (21 days). Mechanical and cold sensitivity as well as locomotion/motivation were tested by the von Frey, acetone, and wheel-running tests, respectively. Additionally, sensory-nerve-action-potential (SNAP) amplitude by caudal-nerve electrical stimulation, electronic microscopy of the sciatic nerve, and inflammatory-protein quantification in DRG and the spinal cord were measured. Interestingly, a higher concentration of curcumin was observed in the spinal cord with the Meriva diet than the curcumin diet. Our results showed that paclitaxel-induced mechanical hypersensitivity was partially prevented by the curcumin diet but completely prevented by Meriva. Both the urcumin diet and the Meriva diet completely prevented cold hypersensitivity, the reduction in SNAP amplitude and reduced mitochondrial pathology in sciatic nerves observed in paclitaxel-treated mice. Paclitaxel-induced inflammation in the spinal cord was also prevented by the Meriva diet. In addition, an increase in α7 nAChRs mRNA, known for its anti-inflammatory effects, was also observed in the spinal cord with the Meriva diet in paclitaxel-treated mice. The use of the α7 nAChR antagonist and α7 nAChR KO mice showed, for the first time in vivo, that the anti-inflammatory effects of curcumin in peripheral neuropathy were mediated by these receptors. The results presented in this study represent an important advance in the understanding of the mechanism of action of curcumin in vivo. Taken together, our results show the therapeutic potential of curcumin in preventing the development of PIPN and further confirms the role of α7 nAChRs in the anti-inflammatory effects of curcumin.

## 1. Introduction

Chemotherapeutic agents used in cancer therapy can cause peripheral neuropathy due to their neurotoxicity [1]. It is estimated that 20%–85% of people undergoing chemotherapy develop peripheral neuropathy, resulting in cancer patients stopping chemotherapy earlier [2,3]. The intensity of chemotherapy-induced peripheral neuropathy (CIPN) differs from person to person. In some cases, the severity of the symptoms may lead patients to reduce or even temporarily stop the treatment in order to relieve their suffering. CIPN can, however, persist long after chemotherapy is stopped [4]. Among the chemotherapeutic agents used to treat solid tumors, paclitaxel is one of the most widely administered. Multiple mechanisms have been reported to mediate the neurotoxicity of paclitaxel, such as local degeneration of distal axon, changes in mitochondrial ultrastructure and transport, increased neuroinflammation, and oxidative stress in the spinal cord and dorsal-root ganglia (DRG) [5,6,7].

The symptoms described by patients are chronic pain, loss of sensation and allodynia in the feet and hands according to a “stocking and glove” distribution, as all of which significantly impact their quality of life [8]. Currently, there is no curative or preventative treatment for paclitaxel-induced peripheral neuropathy (PIPN). Only symptom-relieving treatments are currently available for cancer patients.

Curcumin, a polyphenol extracted from the root of *Curcuma longa*, has been shown to improve functional recovery in many animal models of peripheral neuropathy [9]. For example, the effect of curcumin has been studied in other models of CIPN induced by vincristine, oxaliplatin, and cisplatin. In all these models, curcumin significantly attenuated mechanical and thermal hypersensitivity. The beneficial effects of curcumin have been attributed to an increase in antioxidant enzymes (such as SOD, GPx, catalase, and GSH) and a decrease in NO and lipoperoxidation [10,11]. Indeed, curcumin is considered as a powerful antioxidant, especially due to the presence of the conjugated β-diketone function in its structure [12]. In addition, curcumin has been shown to reduce oxidative stress and inflammation through, among other things, activation of the Nrf2 pathway [13,14,15]. Curcumin also inhibits inflammatory cytokines, such as interleukins (IL-1 and IL-6), TNF-α, and chemokines as well as MAP kinase and NF-κB pathways [13,16]. Recently, it has been proposed that this anti-inflammatory action might involve the activation of the alpha7 nicotinic-acetylcholine receptor (α7 nAChR), whose anti-inflammatory properties are well-described [17]. Indeed, we have shown that curcumin is a positive allosteric modulator (PAM) of α7 nAChRs [18].

However, one of the major problems with the use of curcumin in therapeutics is its low bioavailability, which has been described in rodents as well as humans [19,20,21]. Thus, different delivery approaches and formulations of curcumin have been developed to improve bioavailability and the access of curcumin for the target organs [9]. Meriva is a curcumin-phosphatidylcholine complex composed of 20% of a mixture of curcuminoids (75% curcumin, 15% demethoxycurcumin, and 10% bis-demethoxycurcumin), 40% of phosphatidylcholine, and 40% of microcrystalline cellulose. It has been shown in humans that this formulation significantly improves the plasma concentration of curcumin [22]. In addition, clinical trials have already been conducted with Meriva in the context of chronic inflammatory diseases [23,24].

In this study, we investigated, for the first time, the therapeutic effect of curcumin in the treatment of PIPN, using either unformulated curcumin or Meriva. For this purpose, we tested the efficacy of 1.5% dietary curcumin and Meriva in preventing the development of PIPN in mice. Multiple aspects of PIPN were evaluated. Mechanical and cold hypersensitivity were assessed by the von Frey and acetone tests, respectively, and locomotion by the wheel-running test. Additionally, sensory-nerve-action-potential (SNAP) amplitude, histological parameters and inflammatory markers were measured. Finally, using pharmacological and genetic approaches, we examined the role of α7 nAChRs in curcumin’s beneficial effects in our PIPN mouse model.

## 2. Materials and Methods

### 2.1. Animals

The animal experiments presented in this article were approved by the Association for Assessment and Accreditation of Laboratory Animal Care (AALAC) at Virginia Commonwealth University (VCU, Richmond, VA, USA), and protocol #AM10142 was specifically approved by the Institutional Animal Care and Use Committee at VCU. Of the 96 mice used in this study, 24 were used to assess the bioavailability and tolerability of the curcumin and Meriva diets (*n* = 8/group), and 72 were used to assess the effects of the curcumin and Meriva diets on paclitaxel-induced peripheral neuropathy (*n* = 12/group). The mice used in these experiments were purchased from Jackson Laboratory (Bar Harbor, ME, USA). They were C57BL/6J mice (male and female), aged 12 weeks. Mice null for α7 (on a C57BL/6J background) were purchased from Jackson Laboratory. They were then mated with wild-type (WT) mice to obtain heterozygous mice. For experiments in this study, heterozygous mice were backcrossed for ≥15 generations to obtain knockout (KO) and WT mice. Mice were maintained in an enriched environment with a dark/light cycle of 12 h and a temperature of 22 °C. Mice were housed 4 per cage and had ad libitum access to regular food, with water and either a curcumin 1.5% diet or a Meriva 1.5% diet. The different foods were made by Envigo (a global 18% protein-chow diet; Envigo Teklad) supplemented with 1.5% curcumin or Meriva, depending on the treatment. The behavioral experiments on the mice were performed during the light cycle. Every effort was made to ensure optimal welfare conditions before, during, and after each experiment, and the mice were observed daily for general condition. The weight of the mice was measured every other day. Every effort was made to reduce the number of animals used. The size of the animal groups for the experiments was based on data from previous studies. Observers of the behavioral tests were not aware of the treatment of the animals. Baseline tests (BL) were performed at the beginning of each behavioral test and the mice were randomly assigned to their groups.

### 2.2. Drugs

Seven days prior to the first paclitaxel injection, the standard laboratory chow was replaced with either a control chow composed of 18% protein (Envigo Teklad), or an 18% protein chow diet completed with 1.5% curcumin (Pfaltz & Bauer (Waterbury, CT, USA), 95% diferuloylmethane), or an 18% protein chow diet completed with 1.5% Meriva (Indena, Italy) in pellet form. The mice were maintained on their respective diets during the 4 weeks of the experiment. This protocol was based on previous studies [25] and with our bioavailability and tolerability data (Figure 1).

Paclitaxel (Athenex, NDC 70860-200-50, Richmond, VA, USA) was procured from VCU Health Pharmacy. Paclitaxel was dissolved in a mixture of kolliphor (Sigma-Aldrich, St. Louis, MO, USA), ethanol (Sigma-Aldrich), and distilled water (mixture proportion 1:1:18). Intraperitoneal injections of paclitaxel were performed every other day, with four administrations in total, at a dose of 8 mg/kg. Control mice received the vehicle (1:1:18, ethanol, kolliphor, and distilled water) at a volume of 10 mL/kg, i.p. and followed the same injection schedule.

The selective α7 nAChR antagonist, methyllycaconitine citrate (MLA), was purchased from Tocris Biosciences (Minneapolis, MN, USA). Methyllycaconitine citrate was dissolved in saline solution (0.9% sodium chloride) and prepared in a volume of 20 μL at a dosage of 10 μg. The intrathecally injection of MLA or vehicle was performed with a Hamilton syringe 30 min before the behavioral tests.

The reagents for HPLC-MS/MS were: deionized water (Fisher Scientific, Pittsburgh, PA, USA), Methanol (Fisher Scientific, Pittsburgh, PA, USA), formic acid (Fisher Scientific, Pittsburgh, PA, USA), curcumin (Cayman Chemical, Ann Arbor, MI, USA), and curcumin-d_6_ (Cayman Chemical, Ann Arbor, MI, USA).

### 2.3. Quantitation of Curcumin by High-Performance-Liquid-Chromatography Tandem-Mass Spectrometry (HPLC-MS/MS)

Curcumin was analyzed in the nervous tissues and brain homogenate (1:3 tissue:deionized water) samples, along with blood and brain homogenate calibrators as controls, using a previously described method [26]. In brief, freshly prepared seven-point calibration curves with a range of 5 ng/mL to 500 ng/mL (whole blood) or ng/g (brain tissue homogenate) curcumin (Cayman Chemical, Ann Arbor, MI, USA), a drug-free control (negative control) containing only the cucumin-d6 (Cayman Chemical), the internal standard (ISTD), and a double that contained neither curcumin or internal standard, were analyzed with each batch of samples. Curcumin was extracted from the blood and brain homogenate (1:3 tissue:water) calibrators, controls, and samples by adding 100 ng/mL or ng/g of the internal standard to each calibrator, control, or sample and then vortex-mixing for 30 s. Theses samples were then deproteinized by adding 800 μL of a 0.1% formic acid in methanol solution. They were then mixed for 5 min and centrifuged for 10 min. The supernatant was transferred to a clean test tube before being dried down under nitrogen. The samples were then reconstituted with 50:50 solution of water:methanol and transferred to auto-sampler vials for high-performance liquid-chromatography-tandem mass-spectrometry (HPLC-MS/MS) analysis. Quantitation of curcumin was accomplished using a Shimadzu Prominence/Nexera LC system attached to an 8050 mass spectrometer controlled by LabSolutions software (Shimadzu Corp., Kyoto, Japan). Chromatographic separation was performed on a Restek Raptor Biphenyl (2.1 × 50 mm id × 2.7 µm) column (Bellefonte, PA, USA) with the following mobile phases: (A) 0.1% formic acid in water and (B) methanol. A binary gradient was used with a flow rate of 1 mL/min and the following programming: form 0.00–1.00 min mobile phase B was set at 50% B. Mobile phase B was then increased to 80% B for 1.0 min, held for 0.3 min, and then returned to 50%. The total run time was 3.0 min. The following transition ions (*m*/*z*) were monitored in multiple reaction monitoring (MRM) in negative ion mode (Collision Energy): curcumin 367 > 134 (32); 367 > 148 (18) and curcumin-d6 373 > 134 (32); 373 > 151 (18). The lens 1 Voltage and lens 3 Voltage were set to 18 V and 15 V, respectively. The concentration of each calibrator was determined within ±15% of the expected value. The linear-regression-correlation coefficients (r2) for all calibration curves were ≥0.995 or better. Concentrations of curcumin were determined by a linear-regression plot based on the ratio of calibrator peak areas.

### 2.4. Mechanical Sensitivity: Von Frey Filaments Test

Mechanical hypersensitivity was determined using the von Frey filaments test as previously reported [27].

### 2.5. Cold Sensitivity: Acetone Test

Cold hypersensitivity was assessed via the acetone test as previously reported [28].

### 2.6. Voluntary Wheel-Running Activity

Voluntary wheel running was determined by measuring the distance traveled using the wheel-running test as previously reported [29].

### 2.7. Measurement of Caudal Nerve Conduction

At 21 days after paclitaxel injection, electrophysiological studies were conducted as previously described [7]. Briefly, dorsal caudal nerve conduction was recorded in the tail by stimulation and recording-needle electrodes connected to the PowerLab/26T device (ADInstruments, Colorado Springs, CO, USA). For signal recording, the recording-needle electrodes were placed at the base of the tail and the stimulation-needle electrodes at the end of the tail. The amplitude as well as the latency of the compound-sensory-action potential were recorded.

### 2.8. Quantification of Intra-Epidermal Nerve Fibers (IENFs) by Immunohistochemistry

At 21 days after the paclitaxel injection, IENF staining in mice paws was conducted as previously described by Toma et al. in 2017, with modifications [30]. The hind-paw skin was finely dissected and placed in 4% paraformaldehyde at 4 °C overnight. Samples were included in OCT, and 25 μm sections were made with a cryostat. The skin sections were dried and then immersed in an acetone bath, rinsed with PBS, and incubated in blocking solution (5% normal goat serum and 0.3% Triton X-100 in PBS) for 30 min at RT. Then, the skin sections were incubated with a solution of PGP9.5 primary antibody (1:200 dilution; Fitzgerald—Cat. # 70R-30722, Boston, MA, USA) at 4 °C O/N. After rinsing with PBS, the skin sections were incubated in a secondary antibody solution of goat IgG anti-rabbit (H + L) Alexa Fluor 488-conjugated (1:300 dilution; Life Technologies—Cat. # A11034, Carlsbad CA, USA) for 90 min at RT. Sections were mounted in Vectashield (Vector Laboratories, Burlingame, CA, USA) and observed using a Zeiss Axio Imager A1 fluorescence microscope, with 63× oil magnification (Carl Zeiss, AG, Jena, Germany). Observation and counting of the IEFNs was done in a blinded manner. Nerve-fiber density is expressed as fibers/mm.

### 2.9. Electronic Microscopy of Sciatic Nerve

After euthanasia of the mice at D22, the sciatic nerves were harvested and fixed in 2.5% glutaraldehyde in a sodium cacodylate buffer at 4 °C O/N. On the day after, the samples were rinsed and post-fixed in 1% osmium tetroxide in a 100 mM cacodylate buffer at pH 7.3. Samples were then dehydrated and embedded in epoxy resin. Ultrathin longitudinal sections were collected on mesh copper grids and stained with uranyl acetate and lead citrate. Sections were then examined using a transmission-electron microscope (JEOL JEM-1230) at 100 kV. The axonal and mitochondrial structures were assessed by a single, blinded examiner.

### 2.10. Inflammatory-Marker Analysis by Multiplex Assay

After euthanasia of the mice at D22, the dorsal-root ganglia (DRG) and spinal cord were collected. The preparation of the samples was previously described in [7]. Measurements of the cytokine-expression levels present in the homogenates of DRG and the spinal cord were performed with a Bio-Plex assay plate (Bio-Plex Pro Mouse Cytokine 23-plex Assay, Bio-Rad, CA, USA). The protocol used for the assay was in accordance with the instructions of the manufacturer. The plate was read by the MAGPIX multiplex instrument with the associated software (Bio-Rad, Hercules, CA, USA).

### 2.11. Statistical Tests and Analysis

Statistical analyses of this study were performed with Graphpad statistical software (GraphPad Software, Inc., La Jolla, CA, USA). Statistical data are expressed as mean ± standard error of the mean (SEM). Equality of variance and normality were checked by Brown–Forsythe and Shapiro–Wilk tests, respectively. Data were then compared using one- or two-way analysis of variance (ANOVA), with repeated measures, followed by Tukey’s or Sidak’s post hoc comparison test. Differences were considered significant when *p* was <0.05, <0.01, or <0.001.

## 3. Results

### 3.1. Study of Bioavailability and Tolerability of Curcumin- and Meriva-Enriched Diets

An initial study was conducted on a cohort of untreated naive mice to determine whether the 1.5% curcumin and 1.5% Meriva diets would be well tolerated by the animals (*n* = 8 mice/group) over a 2-week period (Figure 1A). Mice were followed daily to assess their general health, and no signs of deterioration were observed. Mice consumed approximately 3.5–5 g of food per day, with no detectable difference between the three diets. For cur-diet, this represents about 52.5–75 mg curcuminoids/day, and for Meriva-diet, this represents about 10.5–15 mg curcuminoids/day. No significant weight loss was observed in the mice treated with either the 1.5% curcumin or the 1.5% Meriva diets (Figure 1B). Plasma, nervous tissues (spinal cord and sciatic nerve), liver, kidney, and spleen concentration of curcumin were measured by mass spectrometry (MS) at day 15 of treatment with the 1.5% curcumin and 1.5% Meriva diets (Figure 1C,D). No significant difference in the concentration of curcumin in the plasma was observed between the two diets (Figure 1C). It is important to note that the Meriva formulation contains one-fifth of the curcumin found in the diet with unformulated curcumin. Thus, if we relate the plasma concentration to the concentration initially present in the diet, the Meriva treatment increased the curcumin-plasma concentrations by five-fold. Statistical analysis (*t*-test) of the tissue concentration data shows a significantly higher sciaticnerve (*p* < 0.05) and liver (*p* < 0.05) concentration in the mice treated with the 1.5% curcumin diet compared to the 1.5% Meriva diet (Figure 1D). In contrast, a higher concentration of curcumin is found in the spinal cord (*p* < 0.05) and in the spleen (*p* < 0.05) with the Meriva diet.

### 3.2. Curcumin- and Meriva-Enriched Diets Reduce Signs of Paclitaxel-Induced Peripheral Neuropathy

We next investigated whether exposure to the curcumin and Meriva diets could prevent various PIPN behavioral, electrophysiological, and morphological outcomes (*n* = 12 animals/group, Figure 2). The impact of paclitaxel with the curcumin and Meriva diets on voluntary wheel-running activity was investigated by the wheel-running test on day 7 (this day having been determinate according to our previous publication) [29]. The overall one-way ANOVA analysis showed a significant difference between groups (*p* = 0.0096; Figure 2B). Specifically, while no significant difference was observed in distance traveled between the three vehicle groups treated with the regular diet (reg-diet/veh), curcumin 1.5% (cur-diet/veh), or Meriva 1.5% (Meriva-diet/veh) (Figure 2B), a significant decrease in distance traveled was observed in the paclitaxel regular-diet group (reg-diet/PAC) compared to the control group (reg-diet/veh) (*p* < 0.05). In contrast, the two groups treated with paclitaxel and curcumin 1.5% (cur-diet/PAC) or paclitaxel and Meriva 1.5% (Meriva-diet/PAC) showed no difference from the control group (reg-diet/veh) (Figure 2B). In addition, Meriva-diet treatment showed a tendency to increase the distance traveled between reg-diet/PAC and Meriva-diet/PAC (*p* = 0.07). These results suggest that curcumin and Meriva can partially prevent the decrease in wheel-running activity produced by paclitaxel.

The von Frey filament test was used to measure the effect of the curcumin and Meriva diets on mechanical sensitivity at different time points (Baseline: BL, Day 7: D7, D14, and D21) (Figure 2C). An overall two-way ANOVA showed an interaction between treatment and time factors (Figure 2C: [F (5.377, 59.15) = 8.419; *p* < 0.0001]). Post-hoc analysis showed no difference in mechanical sensitivity between the three groups of vehicles at the different time points (BL to D21). However, a significant decrease in the paw-withdrawal threshold was observed in the paclitaxel-treated animals at D7 (*p* < 0.001), D14 (*p* < 0.001) and D21 (*p* < 0.001), reflecting paclitaxel-induced mechanical hypersensitivity. Treatment with the curcumin diet at 1.5% partially prevented this hypersensitivity at D14 and D21. Whereas, treatment with the Meriva diet at 1.5% completely prevented the mechanical hypersensitivity induced by paclitaxel to D14 and partially to D7 and D21 (Figure 2C). These results show the therapeutic potential of curcumin and Meriva to partially prevent the development of mechanical hypersensitivity induced by paclitaxel treatment.

The acetone test was used to measure the effect of the curcumin and Meriva diets on cold hypersensitivity at BL, D7, D14, and D21 (Figure 2D). Two-way ANOVA showed an interaction between the two factors of treatment and time (Figure 2D: [F (15, 198) = 5.070; *p* < 0.0001]). More specifically, no difference was observed between the three vehicle groups. However, a significant increase in the time aversive response was observed at D7 (*p* < 0.05), D14 (*p* < 0.05), and D21 (*p* < 0.01) between reg-diet/veh and reg-diet/PAC groups, reflecting that cold hypersensitivity was induced by paclitaxel (Figure 2D). The 1.5% curcumin and 1.5% Meriva diets completely prevented the development of cold hypersensitivity at D7, D14, and D21 (Figure 2D). Overall, these results show the therapeutic potential of curcumin and Meriva to prevent the development of the cold hypersensitivity induced by paclitaxel treatment.

Finally, sensory-nerve-conduction amplitude (Figure 2E) and velocity (SNCV) (Figure 2F) were measured on day 21 (this day having been determined according to our previous publication [28]). Our electrophysiological data show a decrease in sensory nerve compound-action-potential (SNCAP) amplitude (*p* < 0.01) in the paclitaxel-treated group (Figure 2F: one-way ANOVA; F = 4.834; *p* < 0.001). No significant difference was observed between the control group and the cur-diet/PAC and Meriva-diet/PAC groups (Figure 2F). In addition, a significant difference was observed between the cur-diet/PAC (*p* < 0.001) and Meriva-diet/PAC (*p* < 0.05) groups and the paclitaxel-treated group (reg-diet/PAC); Figure 2F. Finally, no significant differences were observed in SNCV between any of the groups (Figure 2E). Taken together, these results shown that curcumin and Meriva prevent the decrease in sensory-nerve-conduction amplitude induced by paclitaxel.

**Figure 2 pharmaceutics-14-01296-f002:**
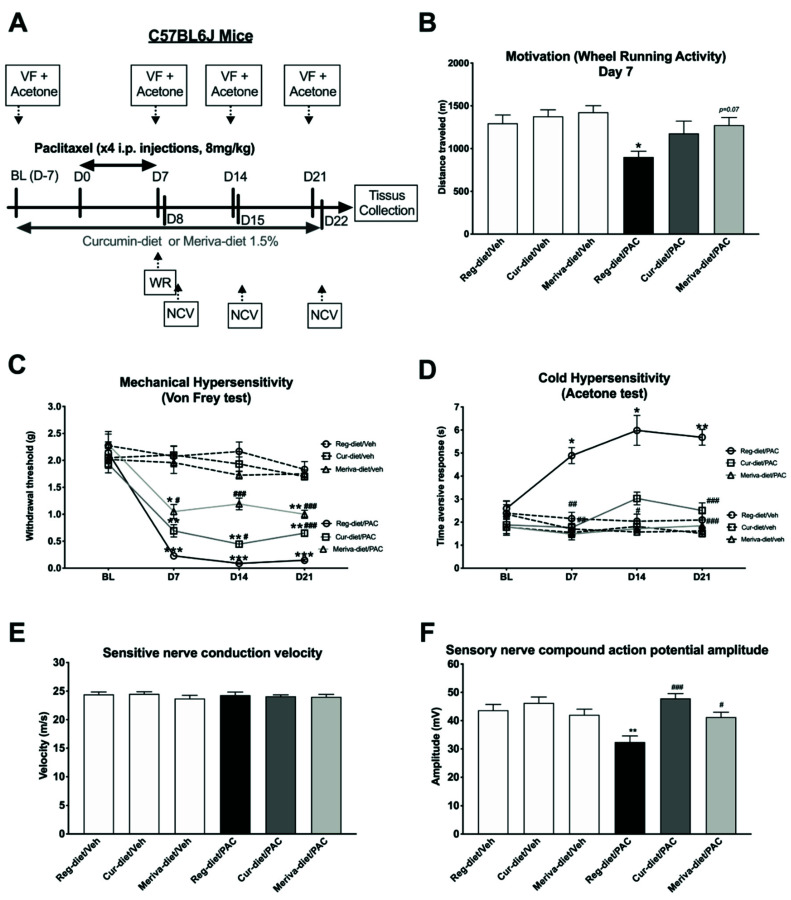
Prevention of PIPN signs with the curcumin and Meriva diets. (**A**) Mice were treated with cur-diet or Meriva-diet for 4 weeks (before, during, and after paclitaxel). (**B**) Motivation and locomotion were evaluated by the wheel-running test at D7. (**C**) Mechanical (von Frey test) and (**D**) cold (acetone test) hypersensitivity were tested at BL, D7, D14, and D21. (**E**) Sensory-nerve-conduction velocity and (**F**) sensory-nerve-compound-action-potential amplitude were measured at D22. Values are expressed as mean ± SEM. *n* = 12/group. Results were compared using two-way ANOVA (**C**,**D**), one-way ANOVA (**B**,**E**,**F**), and post-hoc Tukey’s test (*: *p* < 0.05, **: *p* < 0.01 and ***: *p* < 0.001 vs. reg-diet/Veh; (#: *p* < 0.05, ##: *p* < 0.01 and ###: *p* < 0.001 vs. reg-diet/PAC). BL = baseline; D = day; Cur = curcumin; Reg = regular; PAC = paclitaxel; Veh = vehicle.

### 3.3. Curcumin and Meriva Diets Reduce Mitochondria Damage in Myelinated and Non-Myelinated Nerve Fibers

One of the characteristics of paclitaxel neurotoxicity is the presence of mitochondrial damage in the axons of the peripheral nerves. Thus, a qualitative analysis by electron microscopy was performed on the longitudinal sections of the sciatic nerve of the animals, in order to characterize their mitochondrial morphology (Figure 3). Under control conditions, the double membrane of the mitochondria is easily observed. Moreover, the majority of mitochondria are elongated, reflecting the healthy appearance of these organelles in both myelinated and non-myelinated fibers.

For the paclitaxel-treated group of animals (reg-diet/PAC), there was clear evidence of mitochondrial pathology, as previously described in the literature [31]. We observed a reduced number of elongated mitochondria, the majority of which were ovoid in shape, swollen, and clustered, with the presence of intra-mitochondrial vacuoles and with a disruption of the mitochondrial membranes. Some normal-appearing mitochondria with minimal disruption were also evident. These observations were made for both the myelinated and unmyelinated axons. At this stage (day 21), there appeared to be no significant alterations in the cytoskeleton after paclitaxel treatment (Figure 3).

Treatment with the 1.5% curcumin and 1.5% Meriva diets reduced the mitochondrial pathology induced by paclitaxel (Figure 3). In both the curcumin and Meriva groups, there were more normal-appearing mitochondria than in the paclitaxel-treated group. For example, there was clearly a tendency to have more elongated mitochondria than in the paclitaxel group (reg-diet/PAC), and no clustering of mitochondria was observed. However, some mitochondria still showed pathology, especially regarding the shape of the mitochondria. These results suggest that both the curcumin and Meriva diets are protective against paclitaxel-induced changes.

**Figure 3 pharmaceutics-14-01296-f003:**
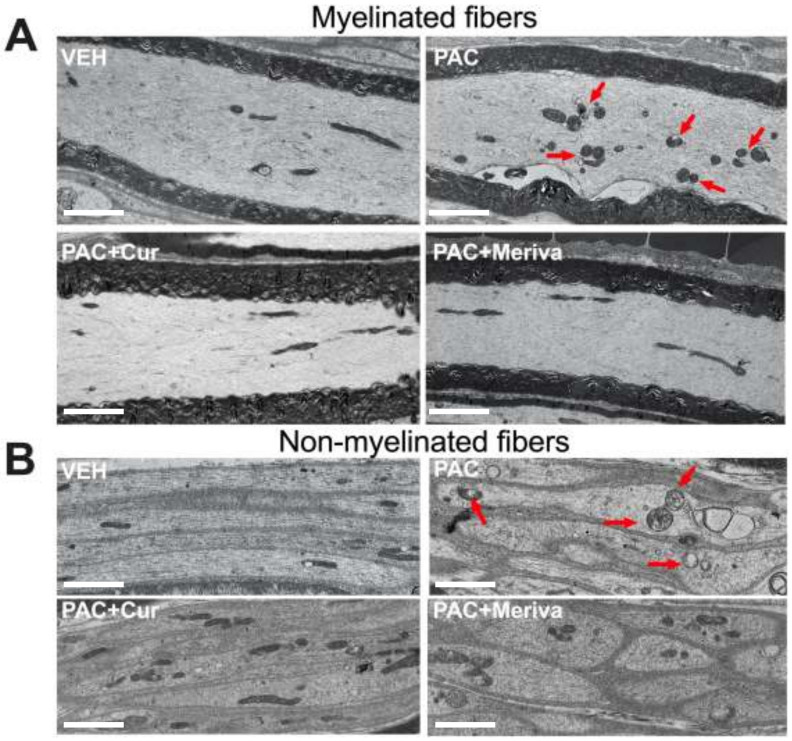
Morphological features of sciatic nerves: transmission-electron-microscopy pictures of sciatic-nerve longitudinal sections. Pictures showing mitochondrial ultrastructure in the control group (VEH), the paclitaxel-treated group (PAC), the paclitaxel + curcumin 1.5%-treated group (PAC + Cur) and the paclitaxel + Meriva 1.5%-treated group (PAC + Meriva), in myelinated (**A**) and non-myelinated fibers (**B**). The red arrows show mitochondria with a pathological aspect. Bar scales: 2.5 μm.

### 3.4. Only Curcumin Prevents the Decrease in Intraepidermal Nerve-Fiber Density

Another important feature of PIPN is the reduction in intraepidermal nerve fibers in the epidermis of the paws. [20]. Thus, to evaluate the impact of paclitaxel and curcumin/Meriva-diet treatment on intraepidermal nerve fibers (IENFs), an analysis of IENF density was performed using immunohistochemistry (Figure 4). No significant differences were observed between the vehicle group (reg--diet/Veh) and either the cur-diet/Veh or Meriva-diet Veh groups. However, at 21 days post-paclitaxel injection, mice treated with paclitaxel (reg-diet/PAC) demonstrated significant reductions in the density of IENFs in comparison to the vehicle group (reg--diet/Veh) (*p* < 0.05; Figure 4A). Interestingly, the curcumin diet appeared to prevent this loss as the IENF density for the cur-diet/PAC group did not differ from the controls (*p* < 0.05) (Figure 4A). No significant differences were observed between the reg-diet/PAC and the Meriva-diet/PAC groups. These data suggest that only the curcumin diet prevents the reduction in IENFs from the paclitaxel treatment.

After comparing the two curcumin preparations in the various aspects of PIPN, we decided to further study the mechanisms of prevention by the Meriva diet, which showed enhanced bioavailability and better results on PIPN signs in the von Frey and wheel-running behavioral tests.

**Figure 4 pharmaceutics-14-01296-f004:**
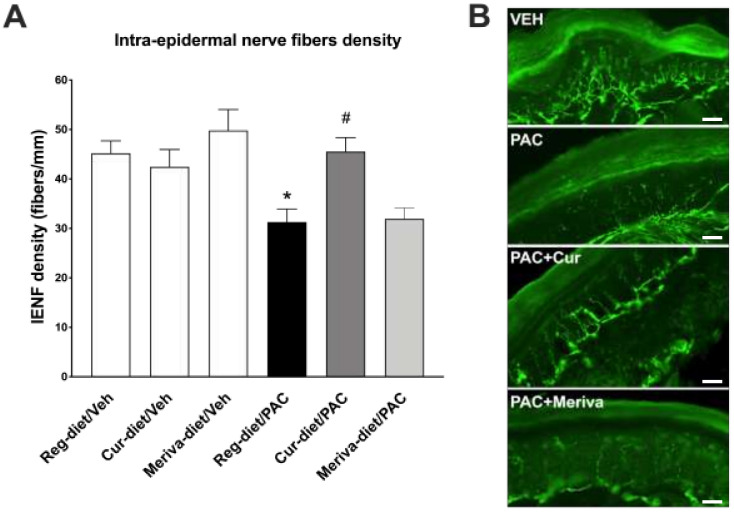
Ur-diet mitigates the reduction in intraepidermal-nerve-fiber (IENF) density at 22 days post-paclitaxel injection. (**A**) Quantification of IENF density in hind-paw skin at day 22 post-treatment. (**B**) Immunostained sections of control (VEH), paclitaxel-treated group (PAC), the paclitaxel + curcumin 1.5%-treated group (PAC + Cur), and the paclitaxel + Meriva 1.5% -treated group (PAC + Meriva), with the hind-paw skin showing the IENFs. Values are expressed as mean ± SEM. *n* = 8/group. Results were compared using two-way ANOVA with post-hoc Tukey’s test and *t*-test (*: *p* < 0.05 vs. vehicle; #: *p* < 0.05 vs. Reg-diet/PAC). Reg = regular; Veh = vehicle; PAC = paclitaxel; Cur = curcumin. Bar scales: 20 μm.

### 3.5. Meriva-Diet Decreases Neuroinflammation in Spinal Cord

An analysis of the levels of the primary inflammatory markers was performed in the DRG and lumbar spinal cord in the multiplex assay at Day 22 after the behavioral testing (Figure 5).

As shown in Figure 5A, the DRG results showed no significant change in the expressions of the different cytokines in the DRG between the control group (WT reg--diet/Veh), the paclitaxel-treated group (WT reg--diet/PAC), and the Meriva-diet paclitaxel-treated group (WT Meriva-diet/PAC) at day 22 (Figure 5A), except for RANTES (WT Meriva-diet/PAC) (*p* < 0.05).

In the lumbar spinal cord (Figure 5B), we observed a significant increase in pro-inflammatory cytokines IL-1α, IL-3, KC as well as a tendency to increase IL-12 (p40) and IL-17α in WT reg-diet/PAC in comparison to the control group. The Meriva diet significantly blocked the increase in these pro-inflammatory cytokines and chemokines. In addition, it decreased the levels of IL-5, G-CSF, GM-CSF, and MIP-1b compared to the paclitaxel group.

Taken together, these results show that on Day 22, paclitaxel-induced inflammation occurred primarily in the lumbar spinal cord and that Meriva-diet reduces this neuroinflammation.

**Figure 5 pharmaceutics-14-01296-f005:**
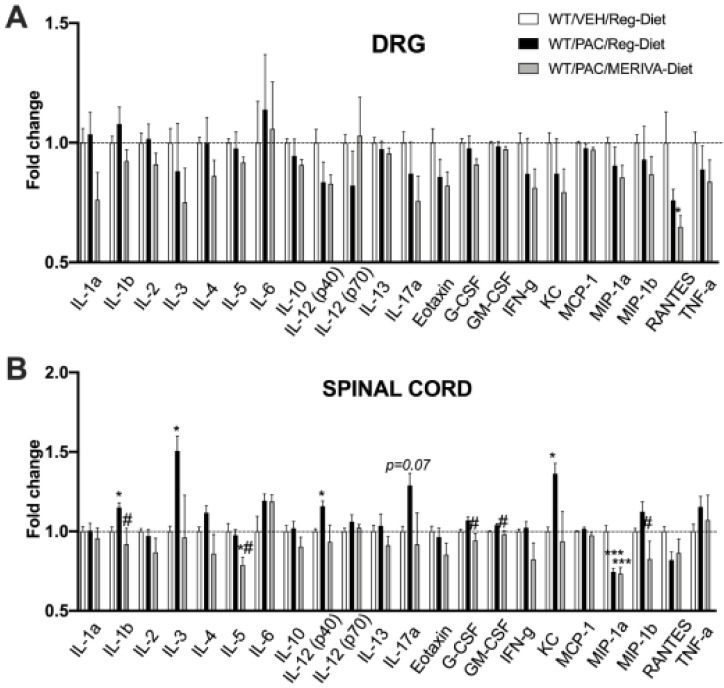
Multiplex-array measurement of inflammation markers in DRG (**A**) and spinal cord (**B**) at 22 days after injection of paclitaxel with or without Meriva-diet. Values are expressed as mean ± SEM. *n* = 6/group. Results were compared using one-way ANOVA and post-hoc Tukey’s test (*: *p* < 0.05 and ***: *p* < 0.001 vs. vehicle group; #: *p* < 0.05 vs. paclitaxel group). WT = wild type; VEH = vehicle; Reg = regular; PAC = paclitaxel.

### 3.6. Meriva-Diet Increases α7 nAchR mRNA Expression in the Spinal Cord

As curcumin has been described to be an α7 nAChR-positive allosteric modulator (PAM) [18], the levels of α7 nAChRs mRNA expression were determined in the DRGs and lumbar spinal cord at day 21 in the different treatment groups. Our results show no significant difference in the expression of α7 nAChRs mRNA in the DRGs of animals in any groups (Figure 6A). However, a significant increase in the expression of α7 nAChRs was observed in the spinal cord in the Meriva-diet/PAC group in comparison to the reg--diet/PAC group (*p* < 0.01) (Figure 6B). No changes in α7 nAChRs mRNA expression were observed in the spinal cord in the other groups (Figure 6B).

Based on the results obtained for the spinal increase in α7 nAChRs expression, an intrathecal injection of an α7 nAChR antagonist (MLA) was performed to determine if the effects of Meriva previously observed with the von Frey test were dependent on the spinal α7 nAChRs. Our results show that 14 days after the first injection of paclitaxel, the Meriva diet partially reversed the mechanical hypersensitivity induced by PAC. Interestingly, the injection of MLA at D15 leads to a decrease in the withdrawal threshold in the Meriva-diet/PAC group compared to the previous day without MLA injection. These findings suggest that the spinal α7 nAChRs mediate the beneficial effects of Meriva in the development of paclitaxel-induced mechanical hypersensitivity.

**Figure 6 pharmaceutics-14-01296-f006:**
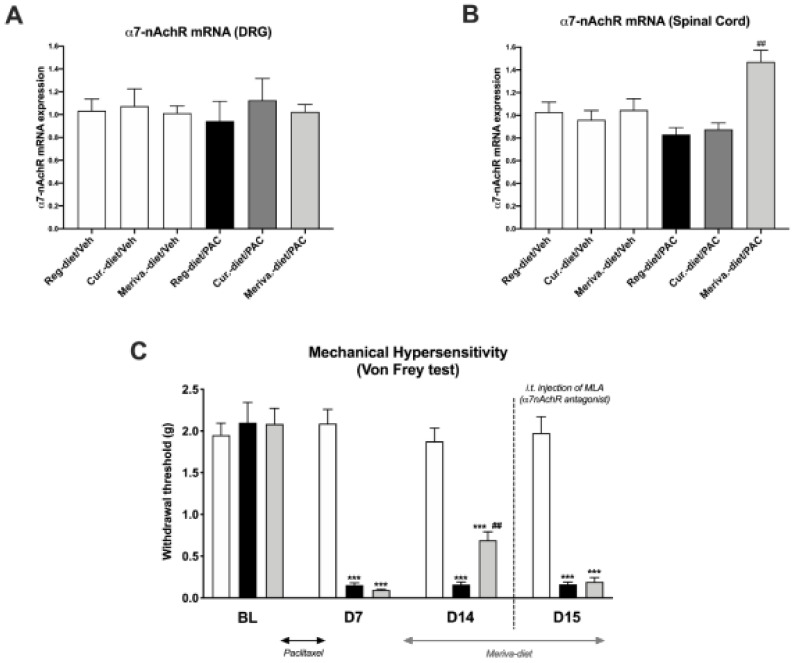
α-7 nicotinic acetylcholine receptor (α7 nAChR) mRNA expression in dorsal-root ganglia (DRG) and spinal cord: α7 nAChR mRNA expression was measured at day 22 after paclitaxel injection in DRG (**A**) and spinal cord (**B**). Values are expressed as mean ± SEM. *n* = 8/group. Results were compared using one-way ANOVA and post-hoc Tukey’s test (***: *p* < 0.001). (**C**) Mechanical hypersensitivity was tested at D7 and D14 after the first injection of paclitaxel with the von Frey test for Meriva-diet and MLA, an α7 nAChR antagonist. Values are expressed as mean ± SEM. *n* = 8/group. Results were compared using one-way ANOVA and post-hoc Tukey’s test (***: *p* < 0.001 vs. vehicle; ##: *p* < 0.05 vs. paclitaxel) Reg = regular; Veh = vehicle; PAC = paclitaxel; Cur = curcumin; D = day; MLA = methyllycaconitine citrate.

The beneficial effect of Meriva in PIPN is largely dependent on α7 nAChRs.

Based on the previous results showing that the α7 nAChR antagonist blocks the effects of Meriva-diet treatment on mechanical hypersensitivity, and a more detailed study was conducted using α7 nAChR global KO mice. Behavioral and electrophysiological measurements were conducted on α7 nAChR KO mice treated with paclitaxel and supplemented with the Meriva diet over 14 days (Figure 7).

Our data from the von Frey test (Figure 7A) show that at baseline (BL) time point α7 nAChR KO mice have a lower threshold of mechanical sensitivity than WT mice (*p* < 0.05). However, at D7 and D14, no significant difference was found between the KO and WT vehicle groups (WT reg-diet/Veh and KO reg-diet/Veh). The Meriva diet did not induce any difference in mechanical sensitivity in the KO and WT vehicle groups (WT Meriva-diet/Veh and KO Meriva-diet/Veh) at D7 and D14. The KO and WT mouse groups treated with paclitaxel and the regular diet showed significantly lower paw-withdrawal thresholds than the controls at D7 (WT reg-diet/PAC: *p* < 0.001; KO reg-diet/PAC *p* < 0.001) and D14 (WT reg-diet/PAC: *p* < 0.001; KO reg-diet/PAC *p* < 0.001), which is indicative of paclitaxel-induced mechanical hypersensitivity in these two groups. As before, the Meriva diet significantly decreased the development of paclitaxel-induced mechanical hypersensitivity in the WT group, compared with the regular diet paclitaxel-treated group at D7 (*p* < 0.05) and D14 (*p* < 0.001), but did not completely block it (D7: *p* < 0.05 and D14: *p* < 0.001 vs WT reg-diet/Veh). However, consistent with the involvement of the α7 nAChRs in paclitaxel-induced toxicity, the preventive effect of the Meriva-diet on paclitaxel-induced mechanical hypersensitivity was largely diminished in the KO mouse group. Thus, these results suggest that the effect of Meriva on mechanical hypersensitivity is dependent on α7 nAChRs.

Cold-sensitivity results (Figure 7B) show that, as before, paclitaxel induces cold hypersensitivity and that Meriva-diet prevents its development in WT mice (D7: WT Meriva-diet/PAC vs WT reg-diet/PAC *p* < 0.001; D14: WT Meriva-diet/PAC vs WT reg-diet/PAC *p* < 0.001). However, the preventive effect of the Meriva diet was not evident in the PAC KO group (D7: Meriva-diet/PAC KO vs. WT reg-diet/Veh *p* < 0.001; D14: Meriva-diet/PAC KO vs. WT reg-diet/Veh *p* < 0.001), again suggesting a critical role for α7 nAChRs in the preventive effect of curcumin.

On day 7, voluntary wheel-running activity (Figure 7C) was significantly decreased in the paclitaxel-treated WT group (WT reg-diet/Veh vs WT reg-diet/PAC: *p* < 0.05). This decrease in voluntary wheel-running activity was prevented by the Meriva-diet (WT Meriva-diet/PAC vs WT reg-diet/PAC: *p* < 0.01). A trend towards a decrease in voluntary wheel-running activity was also observed in KO animals treated with paclitaxel (*p* = 0.07). The same trend was observed in KO animals treated with paclitaxel and the Meriva-diet (*p* = 0.07).

Concerning the electrophysiological data measured at D21 (Figure 7D), a significant decrease in SNCAP amplitude was observed in the paclitaxel-treated WT group (WT reg-diet/Veh vs WT reg-diet/PAC: *p* < 0.05) that was prevented by the Meriva diet (WT Meriva-diet/PAC vs WT reg-diet/PAC: *p* < 0.05). A significant decrease in SNCAP amplitude was also observed in the KO group treated with paclitaxel (KO reg-diet/Veh vs KO reg-diet/PAC: *p* < 0.05). However, the Meriva diet did not prevent this decrease in amplitude in KO mice.

Taken together, these results demonstrate that the effect of Meriva and, thus, curcumin in PIPN is largely dependent on α7 nAChRs.

**Figure 7 pharmaceutics-14-01296-f007:**
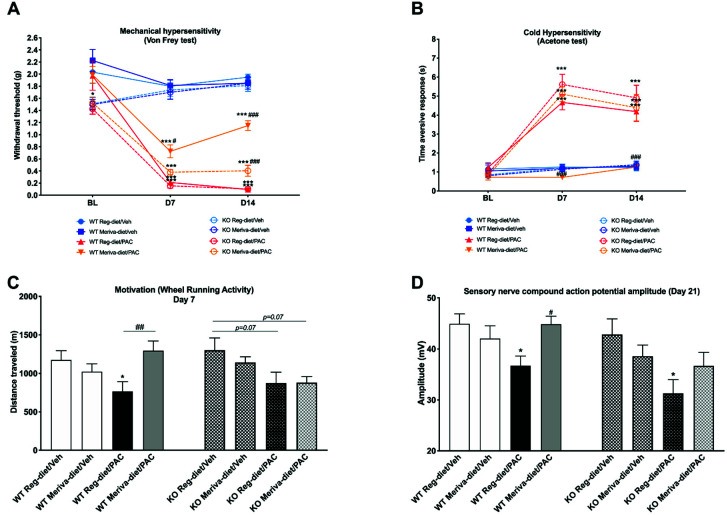
Prevention of PIPN signs by Meriva diets depending on α7 nAChRs: WT and α7 nAChR KO mice were treated with Meriva-diet for 2 weeks (before, during and after paclitaxel). (**A**) Mechanical (von Frey test) and (**B**) cold (acetone test) hypersensitivity were tested at BL, D7, and D14. (**C**) Motivation and locomotion were evaluated by the wheel-running test at D7. (**D**) Sensory-nerve-compound-action-potential amplitude was measured at D21. Values are expressed as mean ± SEM. *n* = 8/group. Results were compared using two-way ANOVA (**A**,**B**) and one-way ANOVA (**C**,**D**) and post-hoc Tukey’s test (*: *p* < 0.05 and ***: *p* < 0.001 vs. WT reg-diet/Veh or KO reg-diet/Veh; (#: *p* < 0.05, ##: *p* < 0.01 and ###: *p* < 0.001 vs. WT reg-diet/PAC or KO reg-diet/PAC). WT = wild type; KO = knock out; Veh = vehicle; Reg = regular; PAC = paclitaxel; BL = baseline; D = day.

### 3.7. Anti-Inflammatory Effects of Meriva Are Reduced in α7 nAChR KO Mice

The anti-inflammatory effects of the Meriva diet observed in paclitaxel-treated WT mice (Figure 5) were tested in paclitaxel-treated α7 nAChR KO mice (Table 1).

Our results in Table 1 show that, in the spinal cord, paclitaxel treatment increased the levels of pro-inflammatory cytokines and chemokines IL-1b (*p* < 0.01), IL-3 (*p* < 0. 001), IL-4 (*p* < 0.05), IL-12(p40) (*p* < 0.001), IL-17α (*p* < 0.01), G-CSF (*p* < 0.01), GM-CSF (*p* < 0.001), KC (*p* < 0.001), MIP-1b (*p* < 0.05) and a trend for TNF-α (*p* = 0.07) in both regular and Meriva-diet-treated α7 nAChR KO mice compared to the control (WT mice). A significant reduction in MIP-1α (*p* < 0.001) and RANTES (*p* < 0.05) was also observed (Table 1).

**Table 1 pharmaceutics-14-01296-t001:** Measurement of inflammation markers by multiplex array in spinal cord at 22 days after injection of paclitaxel in α7 nAChR KO mice with or without Meriva diet. Values are expressed as mean ± SEM of percentage change from control. *n* = 6/group. For all inflammatory markers described in Table 1, the Meriva contrasted with the outcomes observed in WT mice (WT:Veh/reg-diet). Results were compared using one-way ANOVA and post-hoc Dunnett’s test. KO = knock out; Reg = regular; PAC = paclitaxel. Green = significant increase; orange = significant decrease.

	Groups (*n* = 6/Group)	% Change from Control	SEM	*p* Value
IL1α	KO/PAC/Reg-diet	2.30	*±3.178*	*0.8348*
	KO/PAC/Meriva-diet	−6.56	*±3.327*	*0.2825*
IL1β	KO/PAC/Reg-diet	14.80	*±3.066*	*0.0038*
	KO/PAC/Meriva-diet	12.80	*±2.069*	*0.0110*
IL3	KO/PAC/Reg-diet	53.10	*±4.133*	*<0.0001*
	KO/PAC/Meriva-diet	50.60	*±3.839*	*<0.0001*
IL2	KO/PAC/Reg-diet	−0.57	*±7.498*	*0.9896*
	KO/PAC/Meriva-diet	−4.39	*±5.485*	*0.5709*
IL4	KO/PAC/Reg-diet	13.50	*±3.478*	*0.0234*
	KO/PAC/Meriva-diet	13.90	*±3.28*	*0.0191*
IL5	KO/PAC/Reg-diet	2.60	*±5.498*	*0.9342*
	KO/PAC/Meriva-diet	−2.19	*±7.057*	*0.9511*
IL6	KO/PAC/Reg-diet	21.40	*±8.614*	*0.2493*
	KO/PAC/Meriva-diet	−1.75	*±11.55*	*0.9887*
IL10	KO/PAC/Reg-diet	−3.85	*±1.082*	*0.4577*
	KO/PAC/Meriva-diet	−3.09	*±1.871*	*0.5921*
IL12(p40)	KO/PAC/Reg-diet	17.60	*±2.311*	*<0.0001*
	KO/PAC/Meriva-diet	15.30	*±1.892*	*0.0001*
IL12(p70)	KO/PAC/Reg-diet	3.00	*±2.646*	*0.7291*
	KO/PAC/Meriva-diet	7.30	*±4.433*	*0.2147*
IL13	KO/PAC/Reg-diet	−1.91	*±4.262*	*0.9366*
	KO/PAC/Meriva-diet	−7.47	*±5.359*	*0.4165*
IL17	KO/PAC/Reg-diet	22.70	*±4.109*	*0.0029*
	KO/PAC/Meriva-diet	27.00	*±4.498*	*0.0006*
Eotaxin	KO/PAC/Reg-diet	−1.98	*±4.039*	*0.9027*
	KO/PAC/Meriva-diet	−6.13	*±3.82*	*0.4192*
G-CSF	KO/PAC/Reg-diet	6.70	*±1.387*	*0.0018*
	KO/PAC/Meriva-diet	5.40	*±0.7903*	*0.0096*
GM-CSF	KO/PAC/Reg-diet	3.90	*±0.583*	*<0.0001*
	KO/PAC/Meriva-diet	4.00	*±0.3372*	*<0.0001*
IFN-g	KO/PAC/Reg-diet	2.30	*±1.767*	*0.6014*
	KO/PAC/Meriva-diet	−0.92	*±2.457*	*0.919*
KC	KO/PAC/Reg-diet	31.40	*±4.992*	*0.0002*
	KO/PAC/Meriva-diet	33.90	*±3.99*	*<0.0001*
MCP-1	KO/PAC/Reg-diet	3.00	*±0.5581*	*0.0018*
	KO/PAC/Meriva-diet	2.20	*±0.4738*	*0.0187*
MIP-1α	KO/PAC/Reg-diet	−24.26	*±1.665*	*<0.0001*
	KO/PAC/Meriva-diet	−24.18	*±2.026*	*<0.0001*
MIP-1β	KO/PAC/Reg-diet	14.40	*±2.169*	*0.0011*
	KO/PAC/Meriva-diet	9.80	*±2.659*	*0.0195*
RANTES	KO/PAC/Reg-diet	−13.91	*±4.046*	*0.0331*
	KO/PAC/Meriva-diet	−23.93	*±3.386*	*0.0006*
TNF-α	KO/PAC/Reg-diet	18.10	*±6.755*	*0.0703*
	KO/PAC/Meriva-diet	17.00	*±5.577*	*0.0899*

## 4. Discussion

Paclitaxel is one of the most widely used therapies for solid tumors. Unfortunately, its use is often associated with the development of peripheral neuropathy and neuropathic pain, sometimes leading patients to reduce doses of the chemo drug or even terminate treatment [4]. Given that there is currently no preventive or curative treatment for paclitaxel-induced neuropathy, the current work interrogated the potential utility of curcumin, a polyphenol extracted from *Curcuma longa*, which is known for its powerful antioxidant and anti-inflammatory effects [9,32]. It has been shown in other models of peripheral neuropathy that curcumin treatment can prevent and reverse the development of nerve damage [9]. However, curcumin has never been tested in the treatment of paclitaxel-induced neuropathy. Thus, in this study we tested the effect of treatment with two diets either enriched with unformulated curcumin or with Meriva in an animal model of PIPN. Our results show that the Meriva diet provided curcumin in the plasma of mice and in the spinal cord. The Meriva diet decreased the signs of PIPN (cold hypersensitivity, mechanics, decreased SNCV, loss of IENFs, and mitochondrial toxicity) in mice more significantly than the curcumin-diet. The effects of Meriva were attributed to a decrease in inflammatory markers, specifically in the spinal cord. Furthermore, our results showed that pharmacological and genetic blockade of α7 nAChR subtypes mediate the beneficial effects of the Meriva diet.

The root of *Curcuma longa* has been used for centuries in medicinal preparations in Asian countries. This root is mainly used in powder form (tumeric powder) and is known for its antioxidant, anti-inflammatory, and antibacterial effects. Over the last 50 years, it has been shown that the beneficial effects of tumeric powder are derived from curcuminoids, and in particular, from curcumin [33]. It has also been shown that despite its potential therapeutic effects, curcumin has a very low bioavailability and a very rapid metabolism [34,35]. Various approaches have been, therefore, developed to improve the bioavailability and stability of curcumin, such as the combined use of piperine or the encapsulation of curcumin [36]. In the field of peripheral neuropathies, local administration approaches have been proposed in cases where the nerve damage is localized [14,37]. However, in the case of PIPN, the use of a systemic treatment is necessary. Unfortunately, the development and use of new curcumin formulation requires many years of testing and validation before clinical use. Therefore, we used an already-characterized formulation of curcumin in this study, for its effects in a mouse model of PIPN. Meriva, a curcumin-phosphatidylcholine complex composed of a 20% of a mixture of curcuminoids (75% curcumin, 15% demethoxycurcumin, and 10% bis-demethoxycurcumin), 40% of phosphatidylcholine, and 40% of microcrystalline cellulose, was tested in our studies. A bioavailability study in humans with Meriva has already been conducted, as along with several clinical trials in the context of inflammatory diseases, allowing for rapid translation to initial clinical studies for PIPN [23,24]. Our bioavailability results in mice showed that at equal doses of 1.5% curcumin and 1.5% Meriva preparations, an equivalent concentration of curcumin was measured in the plasma, even though the curcumin content in the Meriva formulation was one-fifth of that of the unformulated curcumin. We also observed that the distribution of curcumin in nervous tissues differs between the two preparations. A higher concentration of curcumin in the spinal cord was measured with Meriva, while with unformulated curcumin the concentrations were higher in the sciatic nerve. These differences in distribution in the nervous tissue could be due to the nature of the encapsulation of curcumin (Meriva), which increases its bioavailability and could allow a better passage of the blood–brain barrier to the spinal cord. In contrast, unformulated curcumin, which is highly lipophilic, accumulates at greater levels in peripheral tissues and would preferentially integrate into the myelin sheath of the nerves, for which we have previously shown a high affinity [14]. The lower concentration of curcumin in the sciatic nerve of mice with Meriva-diet compared to unformulated curcumin could partly explain the slightly lower prevention of the decrease in sensory nerve conduction amplitude as well as the lack of effect on the decrease in IENF induced by paclitaxel. Thus, the use of higher concentrations of Meriva should be considered for future studies.

It has previously been shown that curcumin can improve the signs of peripheral neuropathy in models of traumatic, metabolic, and genetic injury [9]. This has also been shown in models of CIPN induced by oxaliplatin, cisplatin, and vincristine [10,11,38,39,40]. In these three models of CIPN, curcumin given by gavage or by i.p. injection decreased mechanical, thermal, and cold hypersensitivity in mice. This was coupled with a decrease in oxidative stress and inflammation. However, the mechanism of action has not been studied. In addition, curcumin has never been tested in the treatment of PIPN. Neuropathy induced by paclitaxel is characterized by the development of mechanical and cold hypersensitivity in the extremities [4]. The neurotoxicity of paclitaxel has been described as due to local degeneration of the distal axons, changes in mitochondrial ultrastructure and transport, increased oxidative stress, and neuroinflammation in dorsal-root ganglia (DRG) and the spinal cord [5,6]. Inflammation and, in particular pro-inflammatory mediators such as TNF-α, IL-6 and IL-1β, sensitize nociceptors to mechanical stimulation and lead to mechanical hypersensitivity [41,42]. In PIPN, inflammation appears to be sequential, with onset initially in the DRGs and later in the spinal cord, which would be responsible for the sensitization [28,43,44]. Therefore, treatments that reduce inflammation and oxidative stress, such as curcumin, seem relevant. Thus, in our study, we observe that adding curcumin prevents the development of mechanical and cold hypersensitivity as well as a decrease in the amplitude of the sensory-compound-action potential and mitochondrial toxicity. We observed a greater effect of Meriva on nociceptive behaviors than with unformulated curcumin. In addition, a decrease in neuroinflammation in the spinal cord was observed with Meriva-diet treatment. This decrease in inflammation in the spinal cord is consistent with the curcumin assay, showing a higher concentration in this organ with the Meriva formulation.

Our results suggest that the α7 nAChR subtype may mediate, to a large extent, the protective effects of curcumin in the mouse PIPN model. Meriva-diet increased expression of this nicotinic subunit in the spinal cord but not in the DRGs in paclitaxel-treated mice only. Furthermore, injection of a selective α7 nAChR receptor antagonist intrathecally completely blocked the partial protective effect of Meriva on the development of paclitaxel-induced mechanical hypersensitivity. These results were confirmed by using α7 nAChR KO mice, in which the protective effect of Meriva on the development of mechanical and cold hypersensitivity, as well as the decrease in the SNCAP amplitude, was blocked. In addition, the decrease in neuroinflammation in the spinal cord by Meriva that was observed in WT mice was not observed in α7 nAChR KO mice. These results extend our previous data on the role of α7 nAChRs in the beneficial effect of curcumin in acute-inflammatory-mouse models and with curcumin as a PAM of expressed α7 nAChRs [18]. It is known that activation of α7 nAChRs expressed peripheral immune cells and produces an anti-inflammatory effect [17]. Thus, the activation of α7 nAChRs can suppress peripheral inflammatory activity by inhibiting the production and secretion of cytokines [45,46]. Moreover, the activation of α7 nAChRs, which have a very high permeability to Ca^2+^, could modulate the excitatory and inhibitory component of synaptic transmission in peripheral neurons, due to their presence on glutamatergic and GABAergic nerve terminals [47,48]. Thus, activation of α7 nAChRs appear to alleviate inflammation-induced pain-like behaviors, and may additionally reduce the sensitizing effect of pro-inflammatory mediators on peripheral sensory neurons [17]. Furthermore, the antioxidant effects of curcumin described previously in other models of peripheral neuropathies may also depend on α7 nAChRs, which are involved in the molecular pathway of Nrf2, a transcription factor involved in the synthesis of antioxidant enzymes that is known to be increased by curcumin treatments [49,50].

In addition to its potential for mitigation of CIPN, numerous studies suggest that curcumin represents a promising candidate as an effective anticancer drug to be used alone or in combination with other drugs [51]. However, in vitro or in vivo future studies need to confirm the lack of interaction of Meriva with the anti-tumor effect of paclitaxel.

Several clinical trials have already been conducted in humans in chronic-inflammation and joint-pain pathologies using different formulations of curcumin [23,52,53]. For example, Meriva has shown strong therapeutic potential in inflammatory knee-pain disease, by reducing patients’ pain and inflammation in plasma [23]. Nevertheless, clinical studies in patients diagnosed with peripheral neuropathies are sorely lacking. An observational study on 80 patients treated with chemotherapy showed that Meriva could decrease the side effects observed in the patients as well as their plasma-free radical levels [24]. However, well-designed clinical trials with Meriva are needed.

## 5. Conclusions

In conclusion, this study shows for the first time that curcumin can largely prevent the development of PIPN in rodents. Thus, in this study, significant improvements were observed with curcumin treatment at the histological, electrophysiological, and functional levels. Moreover, for the first time, we show in vivo that the anti-inflammatory and antinociceptive effects of curcumin in peripheral neuropathy are dependent on α7 nAChR receptors in spinal cord. Curcumin, having already been tested in humans in various clinical trials and having no known toxicity, should be the subject of a new study in the treatment of peripheral neuropathies. This study confirms once again the therapeutic potential of novel formulations of curcumin in the treatment and prevention of peripheral neuropathies.

## Figures and Tables

**Figure 1 pharmaceutics-14-01296-f001:**
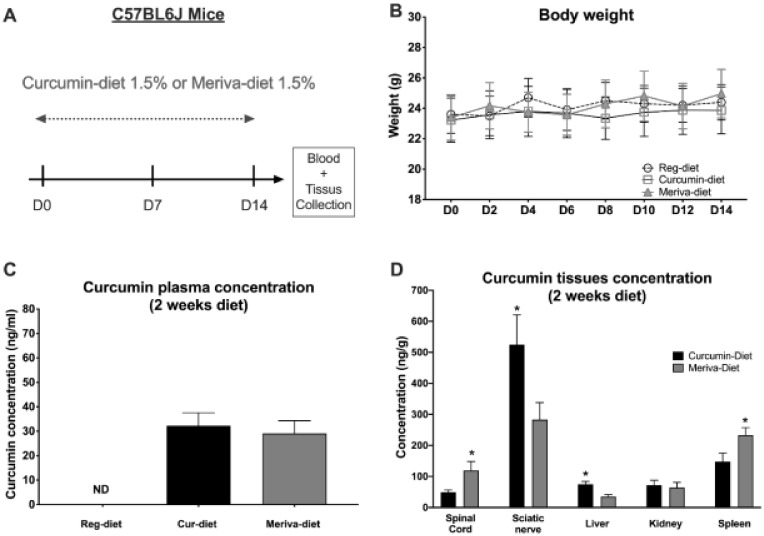
Body weight changes in mice and plasma/tissue levels of the curcumin diet and Meriva diet. (**A**) Experimental design of the curcumin and Meriva diets. (**B**) Body-weight measurement occurred every other day for 14 days. (**C**) Plasma concentration of curcumin after two weeks of the curcumin diet or Meriva diet, measured by mass spectrometry. (**D**) Central and peripheral nervous tissues, liver, kidney, and spleen concentration of curcumin after two weeks of the curcumin diet or Meriva diet, measured by mass spectrometry. Values are expressed as mean ± SEM. *n* = 8/group. Results were compared using two-way ANOVA (**B**) with post-hoc Tukey’s test and *t*-test (**C**,**D**) (*: *p* < 0.05 vs. cur-diet).

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
