# Peer review of "Formulated Curcumin Prevents Paclitaxel-Induced Peripheral Neuropathy through Reduction in Neuroinflammation by Modulation of α7 Nicotinic Acetylcholine Receptors"

_pharmaceutics, 2022, doi:10.3390/pharmaceutics14061296_

Round 1

Reviewer 1 Report

the work presented here is of quality, and well presented

i have some suggestions

1. Please check the wording, for example in line 98 the use of parentheses the protocol #AM10142)

2. The images in figure three, I consider they are not comparative, different approaches are observed (X), can you put similar images of approaches?

3. the study shows important results, but only has a small conclusion, they can increase the conclusion with so many results that they present

4. may include the clinical significance of their findings

Author Response

Response to Reviewers

First, we would like to thank the Reviewers for their constructive comments about our manuscript that were helpful for our revision. A number of changes have been made to the paper; as indexed in this letter and highlighted in yellow in the manuscript. We hope that these changes will be considered as a significant improvement of our manuscript.

Reviewer 1

The work presented here is of quality, and well presented

We thank the reviewer and we appreciate the encouraging comments

I have some suggestions

  1. Please check the wording, for example in line 98 the use of parentheses the protocol #AM10142)

Thanks for pointing out this relevant point. The manuscript has been checked and the typing errors corrected. (Line 102)

  1. The images in figure three, I consider they are not comparative, different approaches are observed (X), can you put similar images of approaches?

The reviewer’s comment is not clear. The images in Figure 3 represent morphological changes in myelinated (A) and non-myelinated fibers (B) in electron microscopy. They are presented in the same scale between the four conditions.

  1. the study shows important results, but only has a small conclusion, they can increase the conclusion with so many results that they present

The conclusion was significantly improved by the addition of several sentences on the striking results obtained in this study and on the clinical potential of curcumin (Lines 637-643).

  1. may include the clinical significance of their findings

A paragraph on the clinical significance and the potential of translational application of our findings in Human has been added in the Discussion section (Lines 626-634).

Reviewer 2 Report

Review of a manuscript “Formulated Curcumin Prevents Paclitaxel-Induced Peripheral Neuropathy through reduction of Neuroinflammation by Modulation of α7 Nicotinic Acetylcholine Receptors” by  Martial CAILLAUD submitted to “Pharmaceutics”.

It is well known that cancer patients treated by chemotherapeutics often are subjected to peripheral neuropathy because of the neurotoxicity of these medications. In patients with severe consequences of such treatment by chemotherapeutics the doctors are often forced to stop the cure. The prevalence of chemotherapy-induced peripheral neuropathy is growing and requires the search of a replacement. The symptoms of peripheral neuropathy may be alleviated by a polyphenol curcumin. The authors studied the effects of curcumin in the treatment of paclitaxel-induced peripheral neuropathy (PIPN).This is an important area of pharmaceutica research and the results od their study will be interesting for the readers of “Pharmaceutics”.

The following corrections and additions should be done.

Introduction

Lines 46- 47: ”The intensity of chemotherapy-induced peripheral neuropathy (CIPN) differs from person to person.” Is there a rational explanation of person to person difference in the intensity of chemotherapy-induced peripheral neuropathy (CIPN)?

Materials and Methods

Line 110 “The different foods were made by Envigo”. This sentence is not informative.What food was supplied to mice?

Results

Figure 1D. The authors should explain the reasons of organ-specific differences in the curcumin concentration shown in Figure 1D. Especially so significant variances between two types of neural tissues, spinal cord and sciatic nerve.

Figures. 

Figure 3 and Fig. 4B. Bar scales should be given.

Discussion

Line 518: ”…a polyphenol extracted from Curcuma longa , that is known for its powerful antioxidant and anti-inflammatory effects [9].” The authors should add here a reference “Phytochemicals as regulators of genes involved in synucleinopathies. Biomolecules, 2021, 11 (5), p. 624”

Conclusion

In conclusion the authors should discuss a potential of translational application of their findings.

Author Response

Response to Reviewers

First, we would like to thank the Reviewers for their constructive comments about our manuscript that were helpful for our revision. A number of changes have been made to the paper; as indexed in this letter and highlighted in yellow in the manuscript. We hope that these changes will be considered as a significant improvement of our manuscript.

Reviewer 2

Review of a manuscript “Formulated Curcumin Prevents Paclitaxel-Induced Peripheral Neuropathy through reduction of Neuroinflammation by Modulation of α7 Nicotinic Acetylcholine Receptors” by  Martial CAILLAUD submitted to “Pharmaceutics”.

It is well known that cancer patients treated by chemotherapeutics often are subjected to peripheral neuropathy because of the neurotoxicity of these medications. In patients with severe consequences of such treatment by chemotherapeutics the doctors are often forced to stop the cure. The prevalence of chemotherapy-induced peripheral neuropathy is growing and requires the search of a replacement. The symptoms of peripheral neuropathy may be alleviated by a polyphenol curcumin. The authors studied the effects of curcumin in the treatment of paclitaxel-induced peripheral neuropathy (PIPN). This is an important area of pharmaceutica research and the results of their study will be interesting for the readers of “Pharmaceutics”.

We thank the reviewer and we appreciate the encouraging comments

The following corrections and additions should be done.

Introduction

Lines 46- 47:” The intensity of chemotherapy-induced peripheral neuropathy (CIPN) differs from person to person.” Is there a rational explanation of person-to-person difference in the intensity of chemotherapy-induced peripheral neuropathy (CIPN)?

Several biological explanations can explain this inter-individual difference. The thresholds of sensitivity to pain are significantly different from one person to another, this may be due to the painful history of the person, to morphological differences and also to the sex of the patient (hormonal protection). In addition, depending on the tumor, the doses and the administration protocol may differ, leading to more or less severe nerve damage.

Materials and Methods

Line 110 “The different foods were made by Envigo”. This sentence is not informative.What food was supplied to mice?

Thanks for pointing out this relevant point. Information on the food has been added to the Methods sections (lines 115-116).

Results

Figure 1D. The authors should explain the reasons of organ-specific differences in the curcumin concentration shown in Figure 1D. Especially so significant variances between two types of neural tissues, spinal cord and sciatic nerve.

A paragraph of the discussion proposes some hypothesis on why we observe a difference of curcumin concentration in the nervous tissue (Lines 559-567).

Figures. 

Figure 3 and Fig. 4B. Bar scales should be given.

Thanks for pointing out this relevant point. Bar scales have been added on Fig.3 and Fig.4B and in the corresponding legend (Lines 364 and 391).

Discussion

Line 518: ”…a polyphenol extracted from Curcuma longa , that is known for its powerful antioxidant and anti-inflammatory effects [9].” The authors should add here a reference “Phytochemicals as regulators of genes involved in synucleinopathies. Biomolecules, 2021, 11 (5), p. 624”

Thanks, this reference has been added in Discussion section (Lines 525)

Conclusion

In conclusion the authors should discuss a potential of translational application of their findings.

The conclusion was significantly improved by the addition of several sentences on the striking results obtained in this study and on the clinical potential of curcumin (Lines 637-643). In addition, a paragraph on the clinical significance and the potential of translational application of our findings in Human has been added in the Discussion section (Lines 626-634).

Reviewer 3 Report

The article is well written and the results are fully analyzed. The results are well discussed. It is only recommended to elaborate on the novelty of work in the abstract and introduction as well as conclusion. The article can be printed in this form.

Author Response

Response to Reviewers

First, we would like to thank the Reviewers for their constructive comments about our manuscript that were helpful for our revision. A number of changes have been made to the paper; as indexed in this letter and highlighted in yellow in the manuscript. We hope that these changes will be considered as a significant improvement of our manuscript.

Reviewer 3

The article is well written and the results are fully analyzed. The results are well discussed. It is only recommended to elaborate on the novelty of work in the abstract and introduction as well as conclusion. The article can be printed in this form.

We thank the reviewer and we appreciate the encouraging comments.

The abstract, introduction and conclusion have been significantly improved to highlight the novelty of this work. (Lines 34-38) (Lines 89 and 92) (Lines 637-643)